# OPERA the Radar Project

**Elena Saltikoff [1,\*], Günther Haase [2]**, **Laurent Delobbe [3], Nicolas Gaussiat [4], Maud Martet [4],**
**Daniel Idziorek [4], Hidde Leijnse [5], Petr Novák [6], Maryna Lukach [3,†] and Klaus Stephan [7]**

[1]  Finnish Meteorological Institute, P.O. BOX 503, FI-00101 Helsinki, Finland
[2]  Swedish Meteorological and Hydrological Institute, 60176 Norrköping, Sweden; gunther.haase@smhi.se
[3]  Royal Meteorological Institute of Belgium, 1180 Bruxelles, Belgium; laurent.delobbe@meteo.be (L.D.);
    M.Lukach@leeds.ac.uk (M.L.)
[4]  Météo-France, 31057 Toulouse, France; Nicolas.gaussiat@meteo.fr (N.G.); maud.martet@meteo.fr (M.M.);
    Daniel.idziorek@meteo.fr (D.I.)
[5]  KNMI, 3730 AE De Bilt, The Netherlands; hidde.leijnse@knmi.nl
[6]  Czech Hydrometeorological Institute, Na Šabatce 17, 143 06 Praha 4, Czech Republic; petr.novak@chmi.cz
[7]  German Weather Service, 63067 Offenbach am Main, Germany; klaus.stephan@dwd.de
\*  Correspondence: elena.saltikoff@fmi.fi; Tel.: +358-29-539-3614
†  Current address: University of Leeds, LS2 9JT Leeds, UK.

**Abstract:** The Operational Program on the Exchange of Weather Radar Information (OPERA) has co-ordinated radar co-operation among national weather services in Europe for more than 20 years. It has introduced its own, manufacturer-independent data model, runs its own data center, and produces Pan-European radar composites. The applications using this data vary from data assimilation to flood warnings and the monitoring of animal migration. It has used several approaches to provide a homogeneous combination of disparate raw data and to indicate the reliability of its products. In particular, if a pixel shows no precipitation, it is important to know if that pixel is dry or if the measurement was missing.

**Keywords:** weather radars; international co-operation; quality control; precipitation

## 1. Introduction

In Europe, the national weather services are responsible for weather observations in their own countries, but also co-operate under the umbrella of the European Meteorological Services Network, EUMETNET. The weather radar work within EUMETNET is co-ordinated by the Operational Program for Exchange of Weather Radar Information (OPERA; www.eumetnet.eu/opera), which was established in 1999. The development and activities of OPERA are described in [1]. OPERA has 30 members operating over 200 radars, mainly in C-band but some in X- and S-bands. The radars come from different hardware and software manufacturers, and some parts were developed in-house. The time span between the oldest and the newest radars in the network is more than 40 years. This makes OPERA, by far, the largest, most heterogeneous body of radar experts anywhere in the world.

Since 2011, the OPERA data center (ODC; also known as "Odyssey") has created Pan-European radar composites every 15 min. This long time-series of data provides an insight to the development of central data processing methods, and also to the gradual improvement and harmonization of the incoming data from over 20 different national weather radar networks.

OPERA produces three different composites with a horizontal resolution of 2 km—maximum reflectivity, rainrate, and hourly accumulation—which are updated every quarter of an hour (on the hour, and at 15, 30, and 45 min past the hour) and are issued approximately 15 min after the start of

data acquisition. It also provides quality-controlled single-site radar data for the numerical weather prediction (NWP) consortia for assimilation.

This paper is organized as follows. In Section 2, we describe the present network and processing methods. In Section 3, we highlight two of the greatest achievements of OPERA. In Section 4, we discuss the reasons for heterogenity of the data and plans for future development.

## 2. Materials and Methods

### 2.1. European Radar Network

In January 2019, OPERA had 30 members. Its metadata database listed 200 operational radars from these members, and 164 of these (from 25 countries) were exchanged regularly. A total of 71 radars were from the 20th century; furthermore, 163 are C-band, 29 are S-band, and 8 are X-band (see Figure 1).

To understand both the variability and common factors in this grouping and to share experiences between members, OPERA has executed several surveys and studies. Results from a survey about maintenance showed large differences in national maintenance policies, and revealed that the primary causes of missing data were not related to the radar itself, but were related to issues with the electricity supplies or telecommunications [2]. A survey about application priorities showed that the most important uses of radar data were for aviation and severe weather warnings. The importance of other applications, such as hydrological applications, NWP assimilation and verification, television, and web had large variability across Europe. These differences of priorities of national meteorological services were the main reasons for the wide variability in measurement schedules observed over Europe. The other main reason was that radars of different generations were used at the same time. Many new radars are more agile than the old ones, allowing more flexibility in scanning. On the other hand, many meteorological services want to collect more samples than before to get the full benefit of their Doppler and dual-polarization capabilities.

The subject of another survey was the use of polarimetric radar variables in Europe. Currently, the largest benefit of dual-polarization (dual-pol) radars is in the quality improvement of other data (reflectivity and Doppler), especially in the removal of unwanted echoes, such as radio interference and wind turbines. Only a limited number of OPERA members are testing dual-pol attenuation correction, and even less have started using this operationally. Many of the new dual-polarization-based quality control algorithms are implemented in the radar signal processor, and are part of the proprietary software. So, they must be applied at the radar site during the time of measurement.

Solar monitoring is a method used to monitor antenna direction. The direction from where microwaves are received from the Sun are compared to the nominal pointing of the antenna and the power from a solar observatory [3,4]. The solar hit detection software in ODC is also able to calculate solar ZDR values [5]. At present (April 2019), polarimetric data (ZDR or TV) is provided by two members (16 radars, in total). In the OPERA data-hub, the solar signals are analyzed in the polar volume data, which helps the members to notice changes in the radar direction. This is a good example of a benefit provided to participating weather services.

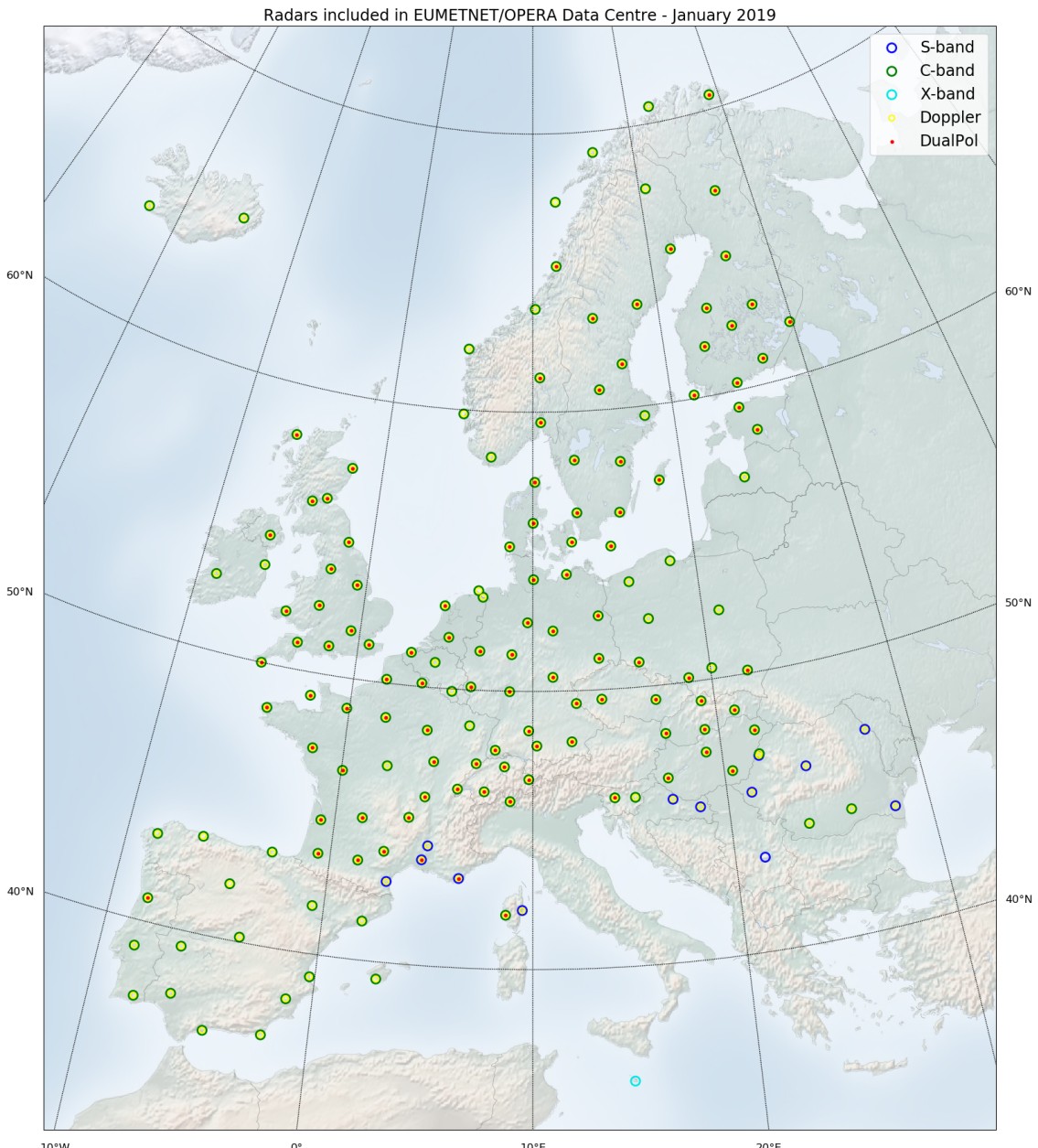

**Figure 1.** Location and type of radars included in the Operational Program for Exchange of Weather Radar Information (OPERA) composite.

## 2.2. Central Processing and Quality Control

In the early years, OPERA was focused on supporting bi-lateral data exchanges. In 2006, OPERA took a first step towards an operational service with the OPERA Pilot Data Hub, which received single-site Cartesian products and national composites from members to create a 4 km resolution pan-European composite. Variability in the quality of the input was surprisingly large, and the re-projection of each of the Cartesian product into the pan-European one led to the loss of resolution and discontinuities. In order to create a better product, it was decided that a data center was needed to receive all the Plan Position Indicator (PPI) scans produced by the European radars, instead of national composites, and, in 2011, the Opera Data Centre (ODC) (later called "Odyssey") was created.

In order to make a composite product that is as homogeneous as possible, the idea (at first) was for each country to send raw polar data scans with only the noise removed and to implement the best quality control methods centrally, in order to apply the same quality control to each radar scan.

A common definition for dry pixels was needed and two terms were defined: Nodata is used to describe that the pixel is out of range or in a blanked sector. Undetect means that the received radar signal is at or below noise level.

The first two central methods to be implemented were an anomaly-removal module [6] and a hit-accumulation filter [7]. The anomaly-removal module uses computer vision methods to identify the patterns (e.g., straight lines or single pixels) that are most often associated with sources not related to weather. The module defines a probability of precipitation that was added to the meta-data as a first quality indicator and rejects the non-precipitation pixels by setting values that exceed a threshold to nodata. The hit-accumulation clutter filter uses a normalized echo count (also called occurrence frequency) calculated each month. In each radar scan, the filter identifies all the pixels for which the normalized echo count exceeds a threshold as residual clutter. A value of 0.6 is typically used.

Two additional methods were implemented in late 2015: Beam blockage correction and a satellite-based filter of residual non-precipitation echoes. The beam blockage correction is described in [8]. The percentage of beam blockage is calculated, in polar coordinates, using a 1 km digital elevation model (GTOPO30) and a geometric propagation model that over-samples the radar beam. For each scan, pre-calculated values are used to correct the reflectivity and are added to the meta-data as a second quality indicator. Reflectivity values in sectors with partial blockage (up to 70%) are corrected and used in the composite products with a lower weight than those in unblocked sectors. Reflectivity values in sectors with blockage exceeding 70% are set to nodata and are not used for compositing.

The satellite filter is based on the EUMETSAT Nowcasting SAF [9] Precipitating Clouds product, which is a probability of precipitation. For each radar pixel where an echo is detected, the 49 satellite pixels surrounding are considered. This smudging of the satellite product allows to take into account the time gap between radar and satellite observations and the effects of parallax in the satellite observations. The maximum probability is used as the third quality index. If the probability of precipitation is 0, the reflectivity is set to undetect.

A quality indicator, defined by the minimum of the three quality indicators presented above, is also added to the radar meta-data. This compound quality indicator is specially needed for NWP assimilation, which heavily relies on high-quality radar observations to produce better forecasts.

Finally, this quality indicator is weighted, according to the distance from the radar and the height of the beam above the ground and used for generating the OPERA composite products maximum reflectivity, rainrate, and hourly accumulations. For each pixel of a composite product, all the single-scan pixels located within the vertical column above are used to calculate the value of the composite field and the associated quality value, following a set of rules.

In the maximum reflectivity composite, the value of the composite reflectivity pixel is the maximum value of all the contributing single-scan reflectivity pixels. The quality index of the composite pixel is also the maximum quality index of the contributing pixels. In the rainrate composite, the value of the composite rainrate pixel is calculated by taking the quality weighted mean reflectivity of all the contributing pixel that are are not nodata or undetect and by applying the Marshall Palmer Z–R relationship (with coefficients $a = 200$ and $b = 1.6$) to obtain a rainfall intensity (mm/h) from the weighted mean reflectivity factor (dBZ). The quality index of the composite pixel is the mean quality index of the contributing pixels. The accumulated precipitation product is simply an integral of the previous four 15 min precipitation intensity products.

At present, the same equation is used to convert reflectivity to precipitation intensity in every weather situation. A more sophisticated method has been studied, but not yet implemented.

In both products, if all contributing pixels are undetect, the pixel value is set to undetect and the quality set to nodata. If all contributing pixels are nodata, the rainrate is set to nodata and the quality set to nodata. If all contributing pixels are either nodata or undetect, the rainrate is set to undetect and the quality index set to nodata.

## 3. Results

### 3.1. Increasing Homogeneity of the European Radar Composites

For quality assessment, OPERA has benefited from co-operation with data users—such as [10,11]—and has also executed its own studies. These studies have made use of the large archive of products we have. Various shortcomings, which are not visible in the 15 min products, can be identified using long-term rainfall accumulations calculated from the archived data. Echoes not related to precipitation, spatial discontinuities or unrealistic spatial variations, and gaps in spatial coverage are examples of shortcomings that can be identified. Next to rainfall accumulations, the frequency of threshold exceedance and the frequency of the special pixel values nodata and undetect are also useful to map. These kinds of long-term statistics can be used as efficient monitoring tools to identify and evaluate the impact of changes in the processing chain.

The first maps show the frequency of nodata for the years 2012 and 2018 (Figure 2). This comparison shows a substantial geographical extension of the network. In 2018, most of Europe is covered, with a data availability exceeding 98%.

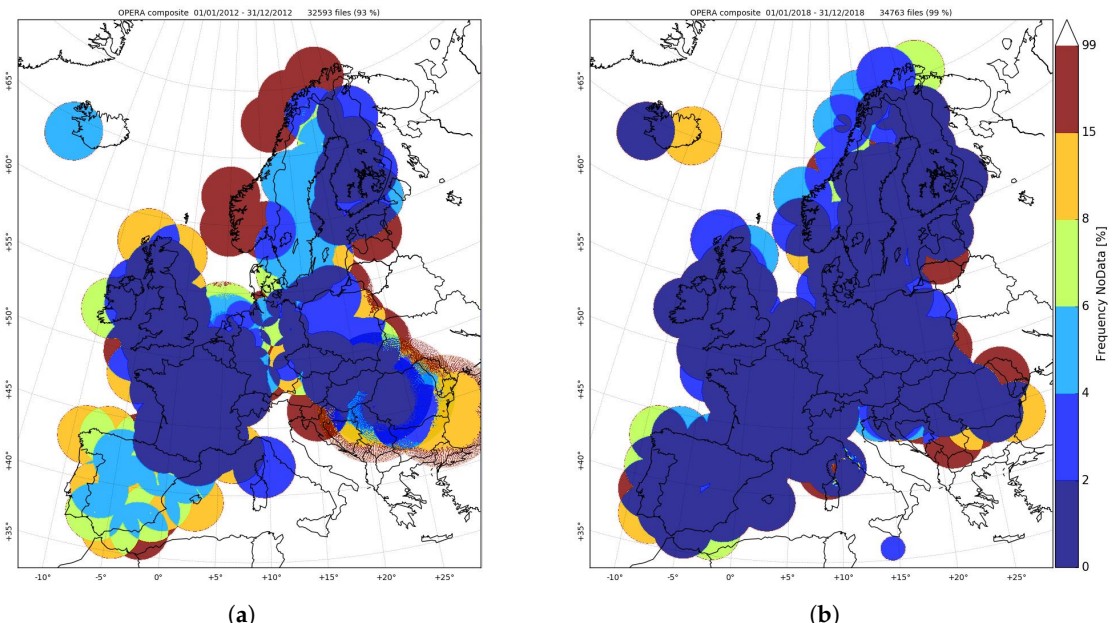

|     |     |
| :-: | :-: |
| (**a**) | (**b**) |

**Figure 2.** Frequency of *nodata* values in (**a**) 2012 and (**b**) 2018.

Figure 3 shows the total rainfall accumulation for 2012 and 2018. Such maps allow for the identification of various artifacts and unrealistic spatial variations. In most areas, an increased homogeneity of the composite was obtained in 2018. Assessment of rainrate composites shows that the increase in homogeneity of composites is due to national improvements, both of the radars themselves and of the data sent to OPERA. Furthermore, the beam blocking correction and satellite-based clutter removal methods, implemented in December 2015, also improved the quality of the composite. This is illustrated in Figure 4, which shows the frequency of undetect values in 2015 and 2016. The undetect value can be interpreted as a "dry pixel". The pixels which are never dry (i.e., blue pixels in the figure) are very likely contaminated by non-meteorological signals, even if these signals are very low in intensity. Comparing 2015 and 2016, we can see the impact of the centralized clutter cancellation methods implemented in the end of 2015.

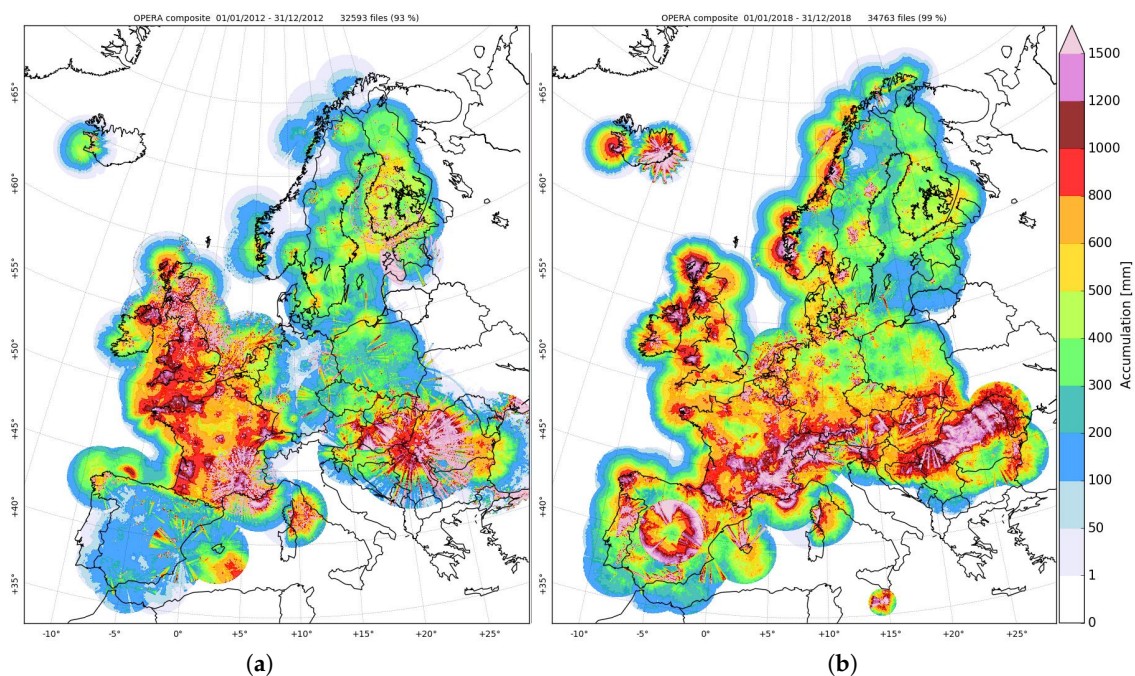

**Figure 3.** Annual precipitation in (**a**) 2012 and (**b**) 2018.

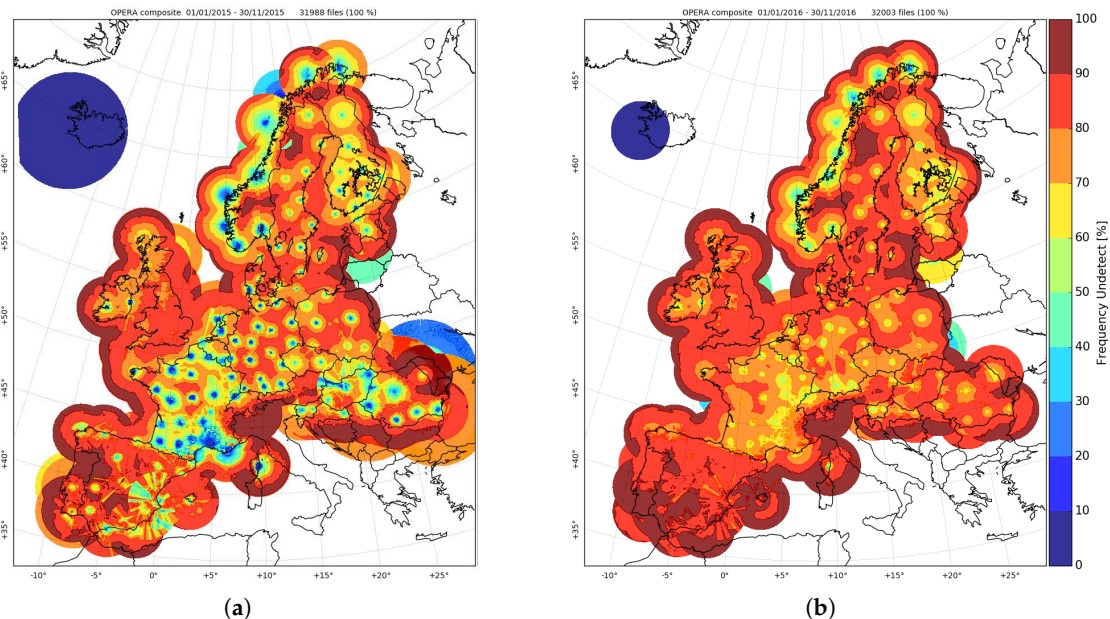

**Figure 4.** Frequency of undetect values in (**a**) January–November 2015 and (**b**) January–November 2016.

It must be noted that some unwanted side effects can be obtained when unwanted echoes, such as sea clutter, are present in a blocked area. In that case, the beam blockage correction will result in an increase of the these unwanted signals. Figure 5 shows the frequency of exceedance of the 0.1 mm/h threshold for 2012 and 2018. It allows identification of areas that are contaminated by ground echoes strong enough to be wrongly interpreted as precipitation. It is easy to identify sectors pointing towards individual radars (e.g., in southern Spain). These signals are related to transmitting devices operating at the same frequency as the radar, which clearly affect the quality of the composite [12]. The comparison of 2012 and 2018 shows a reduction of these disturbances in Eastern Europe, but an increase in other regions.

To quantitatively assess the quality of the OPERA products, an external reference is needed. Hence, the rainrate composites were compared to the accumulated precipitation using the mesoscale analysis system MESAN [13].

Figure 6 (left) shows the number of days in December 2017 where the precipitation intensity in any of the composites is greater than 0.0 mm/h. This data set was, subsequently, used to normalize the precipitation intensity difference (OPERA–MESAN). Figure 6 (right) shows the corresponding average daily bias for December 2017. In general, weather radars underestimated precipitation at the borders of the network due to beam overshooting. This was especially evident at Northern latitudes, but also in the South during winter. On the other hand, precipitation close to a radar is sometimes overestimated, due to remaining clutter. Interferences (e.g., Spain) and potential overcompensation effects of the central processing software (e.g., East Iceland) are clearly visible in Figure 6 (right).

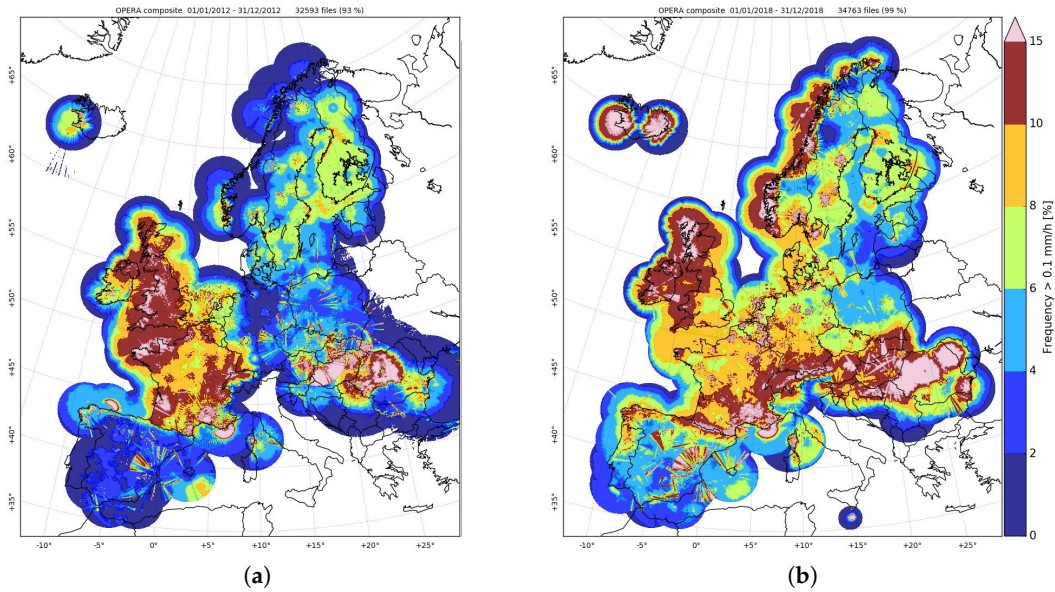

**Figure 5.** Frequency of precipitation over 0.1 mm/h in (**a**) 2012 and (**b**) 2018.

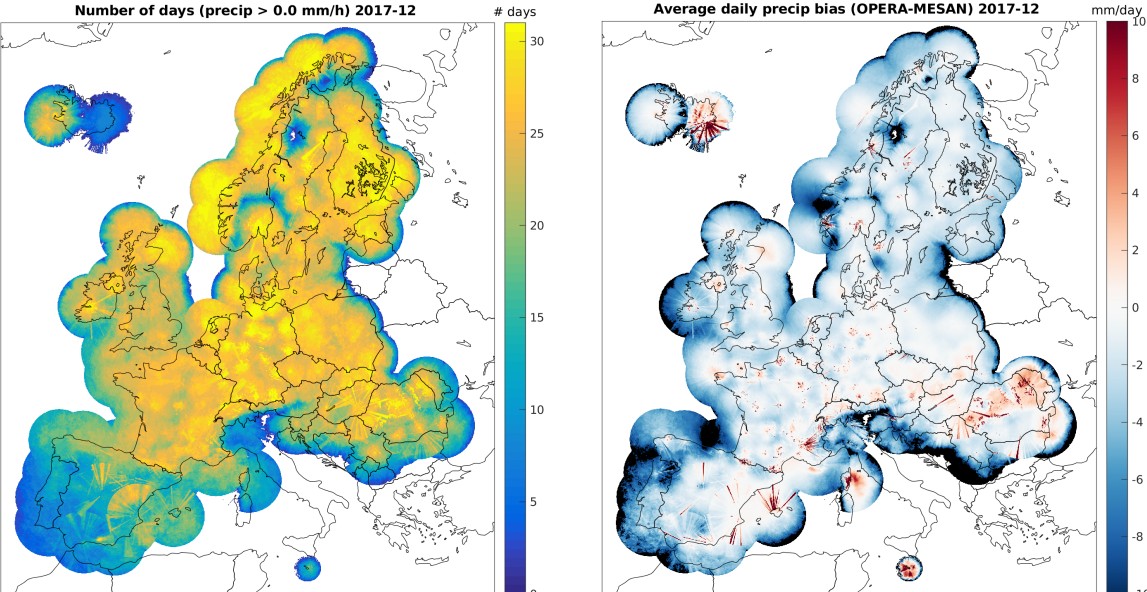

**Figure 6.** Number of days in December 2017 where the precipitation intensity in any of the composites was greater than 0.0 mm/h (**left**). Daily precipitation intensity bias (OPERA–MESAN), normalized by the number of days, where the precipitation intensity in any of the composites was greater than 0.0 mm/h (**right**).

Figure 7 shows the monthly average precipitation bias (OPERA–MESAN) for 2015–2017. Typically, OPERA underestimates precipitation by 1–3 mm/day. This bias was smaller in summer and larger in winter, as beam overshooting is more common in winter. The zero bias in March 2016 looks good, but for the wrong reason. In fact, a single malfunctioning radar (failing noise reduction over two hours) compensated for the general underestimation of precipitation in the OPERA composite domain.

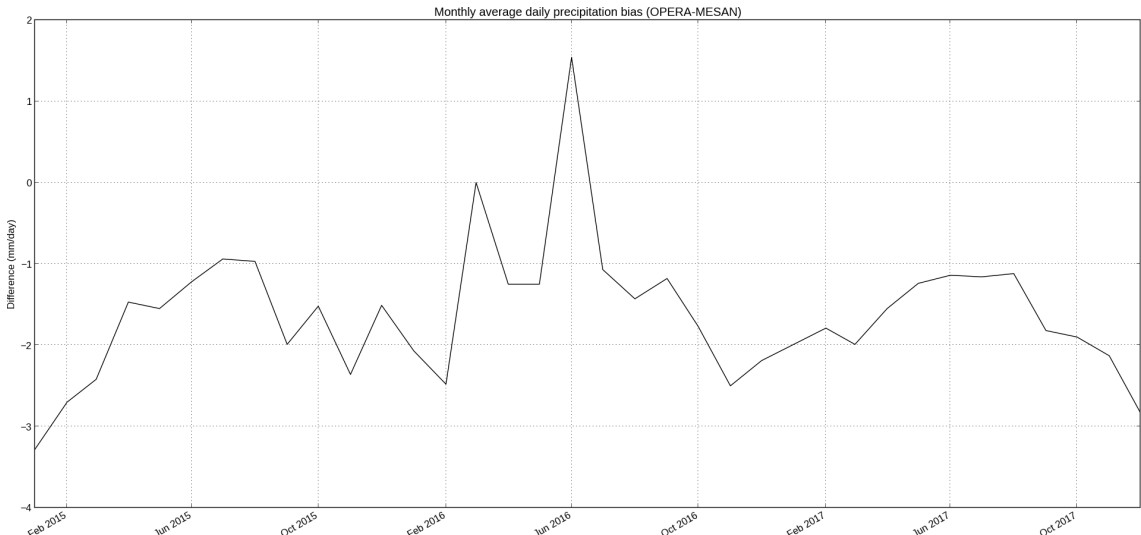

**Figure 7.** Monthly average daily precipitation intensity bias (OPERA–MESAN) over the OPERA composite domain.

A clutter analysis, based on composite pixels with precipitation intensities greater than 0.0 mm/h, confirmed that the monthly average clutter fraction decreased at the end of 2015, which coincides with the implementation of the new clutter filters in the central processing software. This did not affect the average bias of the rain rates, as clutter echoes are often quite weak.

More extensive reports of these studies, as well as a selection of other OPERA studies, are available at the OPERA website under the heading "OPERA Publications" [14].

### 3.2. Common Data Format

During the first phase of OPERA, starting in 1999, the community adopted and promoted use of the BUFR format defined by World Meteorological Organization (WMO). Before that, each radar owner used the native formats of the software manufacturers, or nationally developed versions of these. Introducing a common format was a revolution, which allowed manufacturer-independent joint development, and made the international exchange of radar data much easier.

The second phase (2004–2006) formulated a second-generation information model for use with weather radar data and the Hierarchical Data Format version 5 (HDF5) file format. It was implemented in 2009, and has been updated several times. Version 2.3 was published in 2019. During the 20 years of OPERA, all major weather radar software manufacturers have adopted the OPERA data format and provided conversion software for it. Additionally, the numerical weather prediction model consortia are currently developing tools to read the ODIM format.

Development of a common data model started with a focus on Cartesian radar products (such as CAPPI or MAX); however, since 2013, bi-lateral and multi-lateral exchange between OPERA members has consisted of single-site data in the original polar coordinates.

The information model was designed from the point of view of trying to harmonize all relevant information, independently of radar manufacturer and the organization from which the data originates. While the information model is intended to enable the representation of the data and products presently agreed upon within the framework of EUMETNET OPERA, we also look ahead to future needs and have tried to ensure that they will be appropriately met with this information model. This means that

known products are supported, as are polarization diversity variables and virtually any quality-related information characterizing a given data set. It is also vital to recognize the importance of being able to represent polar (spherical coordinate space) data, and this is accommodated, as well, in a flexible manner.

### 3.3. Data for Users

OPERA produces three different composites [15], which are updated every quarter of an hour (on the hour, and at 15, 30, and 45 min past the hour), issued approximately 15 min after the start of data acquisition. The OPERA composites, covering large parts of Europe with a spatial resolution of 2 km × 2 km (Lambert Azimuthal Equal Area projection), are distributed in the ODIM HDF5 format. The maximum reflectivity product is also shown as an animation on the EUMETNET website.

Odyssey was developed with four user groups in mind: Forecasting and nowcasting, numerical weather prediction (assimilation and verification), civil and military aviation, and hydrology and water management.

For the weather forecasters of national weather services, the products from their national radar network are always the priority. A radar is a mesoscale tool, and, while its strength is superior resolution in time and space, it is most suitable for the observation and analysis of short-lived, quickly-developing mesoscale weather phenomena, such as thunderstorms. For such an analysis, the unavoidable delay of creating a continent-wide composite is too long. However, the composites can be useful for synoptic scale analysis; see Figure 8.

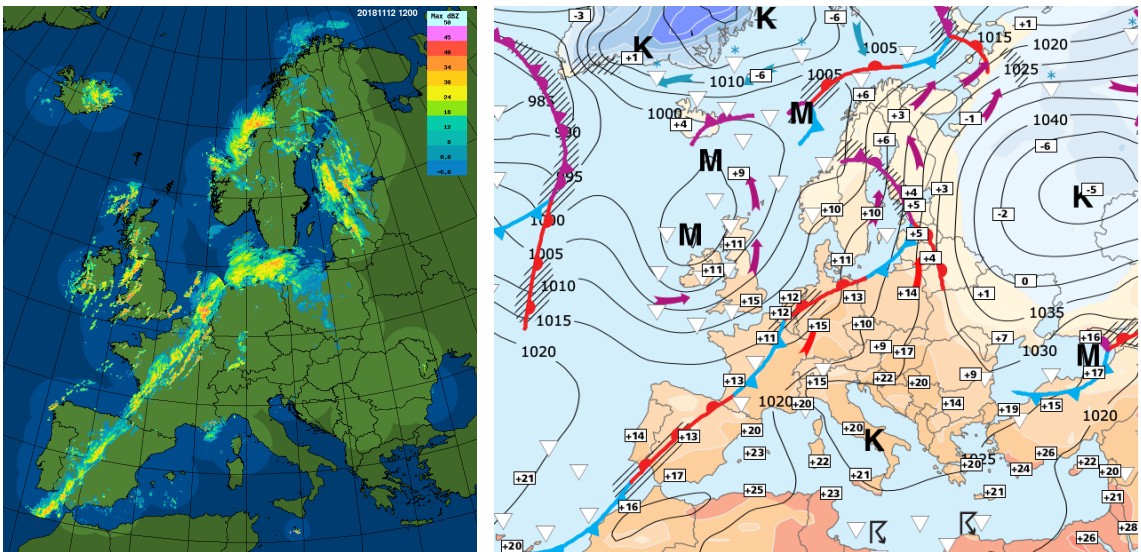

**Figure 8.** OPERA composite at 12 UTC on 12 November 2018 (**left**) and the corresponding weather map (**right**, courtesy of the Finnish Meteorological Institute).

For nowcasting, several systems are in use in Europe. [16] showed that assimilation in rapid-update cycle models outperforms observations-based extrapolation after 2 h. However, there are user groups which are most interested in the first 2 h, especially in the aviation community.

For the assimilation of radar data in numerical weather prediction models, several approaches are used around Europe. Most of the limited area models use radar data in their original radial coordinates in order to take full benefit of the three-dimensional structure of the precipitating system. Global models have used two-dimensional composites. In the European Centre of Medium-Range Weather forecasting (ECMWF), Lopez et al. [17] have used 6-hourly precipitation sums in 4DVAR. They have implemented the use of NCEP Stage IV data from the contiguous United States operationally. This data is a combination of radar and rain gauges. A similar product for Europe is not available, and the quality of OPERA composites alone have not yet been sufficient for such use. The COSMO consortium has tried

to use the rainrate composite for Latent Heat Nudging [18] and experimented with the assimilation of 3-dimensional radar reflectivities with an ensemble Kalman filter [19]. The HARMONIE consortium has used an indirect 3DVAR approach for single-site radar data [20]. Météo France has assimilated radars in AROME, combining 1-dimensional and 3-dimensional variational assimilation [21,22], where 30 French radars have been operationally used (Météo France in-house-product) and 62 radars from OPERA data are planned to be introduced. The LACE consortium has used over 150 radars.

The OPERA composites are also available for commercial use under the rules of ECOMET. This is not a cheap option—the price is the same as it would be to buy all the national data sets used as input—but the ease of obtaining the composite from one place has been indicated as satisfying by a few customers.

Park et al. [23] have used the OPERA composites to assess flash flood hazard induced by heavy rainfalls in European scales, together with the European Flood Awareness System (EFAS). So far, the quantitative accuracy of the OPERA composites alone has not been high enough for their needs, but they have combined the point accuracy of gauge data and the superior resolution of the radar data.

The data collected for weather service purposes has also been used by animal migration researchers [11]. It has been shown that the OPERA network can be used effectively to map continental-scale bird migration events [24].

## 4. Discussion

Definition of a manufacturer-independent data model has been an unprecedented success story. The OPERA data model has been used in international radar data exchange and re-analysis beyond the borders of Europe (e.g., in Australia). The first versions were very accommodating, allowing easy conversion from all the different manufacturer-specific formats; even the identifier of a radar can be given in five ways. Now, as the use of radar data has expanded from exchange between members or organizations to assimilation, verification, and data fusion in various communities, there is more pressure to tighten the use of the model, in order for it to be more homogeneous. The OPERA program has responded to this by allocating a new work package for homogenisation of the metadata.

Each of the participating institutes has defined the use of its radars primarily for its own uses, within the boundary conditions of the hardware and data transfer capacity. Creation of an international composite has resulted in a natural pressure to homogenise the radar network. Sharing best practises in expert meetings has led to the same direction. The level of homogeneity can be considered a great achievement, in view of the heterogeneity of the hardware and measuring circumstances. The different brands and ages of the radars can be compared to those of cars in Europe: If we were to dictate that everyone must run at the same speed, we should force the Ferraris to slow down to Trabant speed, or agree that Trabants can never reach the Ferrari standard.

It is important to note that these changes have happened gradually, and that the OPERA archives contain data from the early years, when the number of non-precipitating echoes was larger and the number of participating radars smaller. The heterogeneity of the time-series can be a risk, if used for climate studies.

Much of the improvement in quality is related to better cancellation of non-precipitating echoes. This is most important for two user groups: Assimilation in numerical weather prediction models, and in extrapolation-based nowcasting. Assimilation of radar reflectivities can compensate for the known spin-up problem of many limited area NWP models: They tend to have too much precipitation at the beginning of the run, and it is valuable if the radars can tell the models where it is not raining. Nowcasting systems calculate motion vectors by comparing two consecutive radar images. Many clutter pixels are permanent and, thus, give false information of zero movement of the precipitating system.

Even with a centralized quality control, the quality of the composites is still not perfect and, most alarmingly, the processing has destroyed some real precipitation data. Meanwhile, several members upgraded their radars and could locally use very advanced signal processing techniques.

Hence, in March 2017, a new agreement was made: Everyone was to send their best possible data, documenting how it was processed, and the unfiltered data (TH) as a separate parameter. This decision has been implemented gradually.

Improvements to the composites which have been studied, but not yet implemented, include the vertical profile of reflectivity, and changing the $Z - R$ relationship, according to water phase (snow or rain).

It has also become obvious that there are disparate user needs: Some applications need advanced quality control and production of complicated products, while others need the data as soon as possible. Hence, from 2019, the OPERA project has focused on development of three separate production lines: One for rapid processing, one for good quality, and one for independent processing (e.g., outside of the OPERA data centers).

**Author Contributions:** E.S. wrote the original draft for Introduction, Sections 2.1 and 3.3 and Discussion. L.D. and G.H. performed the data analysis for Section 3.1 and wrote the original drafts for those paragraphs. G.H. lead the work for Section 3.2. M.M. lead development for Section 2.2. N.G. and D.I. contributed in writing the text of Section 2.2. M.L. developed software for Figures 2–5. P.N. created Figure 1. K.S. wrote about NWP in Section 3.3. All authors participated in the writing, review, and editing of the paper.

**Funding:** This research was partially funded by EUMETNET.

**Acknowledgments:** The authors wish to thank the past and present members of OPERA Expert team for their support and discussions during the development of achievements described in this article.

**Conflicts of Interest:** The authors declare no conflict of interest.

## Abbreviations

The following abbreviations are used in this manuscript:

| | |
|---|---|
| AROME | Application of Research to Operations at Mesoscale |
| BUFR | Binary Universal Form for the Representation of meteorological data |
| CAPPI | Constant Altitude Plan Position Indicator |
| COSMO | Consortium for Small-scale Modeling |
| dBZ | logaritmic unit of radar reflectivity |
| ECMWF | European Centre of Medium-Range Weather forecasting |
| ECOMET | Economic interest grouping of the National Meteorological Services of the European Economic Area |
| EFAS | European Flood Awareness System |
| EUMETNET | European Meteorological Services' Network |
| EUMETSAT | European Organisation for Meteorological Satellites |
| GTOPO30 | Global 30 Arc-Second Elevation–digital elevation model from United States Geological Survey |
| HARMONIE | Hirlam-Aladin Research–(towards) Mesoscale Operational NWP In Europe |
| HDF | Hierarchial Data Format |
| LACE | Regional Centre for Limited Area modelling in Central Europe |
| MAX | Maximum Reflectivity |
| MESAN | Mesoscale analysis system |
| NCEP | National Centers for Environmental Prediction |
| NWP | Numerical Weather Prediction |
| ODC | Opera Data Centre (Odyssey) |
| ODIM | Opera Data Information Model |
| OPERA | Operational Program for Exchange of Weather Radar Information |
| PPI | Plan Position Indicator |
| QI | Quality Index |
| RLAN | Radio Limited Area Network |
| SAF | Satellite Application Facility |
| TH | Total (unfiltered) reflectivity in horizontal polarization |
| TV | Total (unfiltered) reflectivity in vertical polarization |
| WMO | World Meteorological Organization |

Z-R        conversion from radar reflectivity factor to precipitation intensity
ZDR        differential reflectivity
3DVAR    Three-Dimensional Variational Data Assimilation
4DVAR    Four-Dimensional Variational Data Assimilation

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
