# Peer review of "OPERA the Radar Project"

_atmosphere, doi:10.3390/atmos10060320_

Round 1
Reviewer 1 Report
In this paper, the improvements in the OPERA weather radar network between the years 2002 and 2018 are presented. Results presented by authors show a substantial geographical extension of the European network between these years increasing the data homogeneity and reducing the non-data values.
The paper is well organized and the presentation is clear and reasonably concise. The paper presented appears to make a good contribution to the literature in order to facilitate the exchange of weather radar data, and therefore, it deserves attention to be published in the Atmosphere journal.
Author Response
Thank you very much for your encouraging comments!
Reviewer 2 Report
Please see the reviews as attached.

Author Response
Thank you very much for your constructive comments.
After re-organizing the chapters, the Z-R discussion comes after material and methods, so we added the following sentences
"At the moment, the same equation is used to convert reflectivity to precipitation intensity in every weather situation. A more sophisticated method has been studied but not yet implemented."
About the beam blockage, we extended the paragraph.
Old text:
"The beam blockage correction is described in [15]. The percentage of beam blockage is calculated in polar coordinates using a 1 km digital elevation model (GTOPO30) and a geometric propagation model that over-samples the radar beam. For each scan, the pre-calculated values are used to correct the reflectivity and are added to the meta-data as a second quality indicator. Reflectivity values in sectors with partial blockage up to 70% are corrected. Data in sectors with blockage exceeding 70% are set to NODATA."
The new text:
"The beam blockage correction is described in [15]. The percentage of beam blockage is calculated in polar coordinates using a 1 km digital elevation model (GTOPO30) and a geometric propagation model that over-samples the radar beam. For each scan, the pre-calculated values are used to correct the reflectivity and are added to the meta-data as a second quality indicator. Reflectivity values in sectors with partial blockage up to 70% are corrected and used in the composite products with a lower weight than those in unblocked sectors. Reflectivity values in sectors with blockage exceeding 70% are set to NODATA and are not used for compositing"
We wish these additions clarify the processing. The typographical issues have also been corrected.
Reviewer 3 Report
This is an interesting paper that reports on the issues that arise when developing a multi-national radar mosaic. The use of long-term accumulations and other statistics was very interesting and of value to other operators of heterogeneous radar networks.
The structure of the paper could be improved by moving section 3 to the end of the paper and modifying it slightly to form a discussion/conclusion section. Also section 4 seems to fit better at the start of the paper just after the introduction to set the scene.
Author Response
Thank you for your encouraging comments!
We took the original order of sections from the MDPI latex template provided in the Overleaf service, but will modify to more traditional structure as recommended by two reviewers and the editor.